# Spatial Modeling of Extreme Temperature in Northeast Thailand

**Prapawan Senapeng** [1] **, Thanawan Prahadchai** [2] **, Pannarat Guayjarernpanishk** [3] **, Jeong-Soo Park** [2] **and Piyapatr Busababodhin** [1,*]

1   Digital Innovation Research Cluster for Integrated Disaster Management in the Watershed, Mahasarakham University, Maha Sarakham 44150, Thailand; prapawan.s@msu.ac.th
2   Department of Mathematics and Statistics, Chonnam National University, Gwangju 61186, Korea; tanawanp.st@gmail.com (T.P.); jspark@jnu.ac.kr (J.-S.P.)
3   Faculty of Interdisciplinary Studies, Nong Khai Campus, Khon Kaen University, Nong Khai 43000, Thailand; panngu@kku.ac.th
*   Correspondence: piyapatr.b@msu.ac.th; Tel.: +66-92-542-6396

**Abstract:** The objective of the present study was to examine and predict the annual maximum temperature in the northeast of Thailand by using data from 25 stations and employing spatial extreme modeling which is based on max-stable process (MSP) using schlatter's method. We analyzed extreme temperature data using the MSP using latitude, longitude, and altitude variables. Our result showed that the maximum temperature has an increasing trend. The return level estimates of the study areas from both the local generalized extreme value (GEV) model and MSP models show that the Nong Khai, Maha Sarakham, and Khon Kaen stations had higher return levels than the other stations for every return period, whereas Pak Chong Agromet had the lowest return levels. Furthermore, the results showed that MSP modeling is more suitable than point-wise GEV distribution. We realize that the spatial extreme modeling based on MSP provides more precise and robust return levels as well as some indices of the maximum temperatures for both the observation stations and the locations with no observed data. The results of this study are consistent with those of some previous studies. The increasing trend in return levels could affect agriculture and the surrounding environment in northeast Thailand. Spatial extreme modeling can be beneficial in the impact management and vulnerability assessment under extreme event scenarios caused by climate change.

**Keywords:** climate change; generalized extreme value distribution; spatial analysis; return level; max-stable process

## 1. Introduction

Global climate change has become an urgent matter of concern as it threatens our planet and affects lives and economy both directly and indirectly. Direct effects include, among others, the risk of flooding from rising sea levels, while indirect effects include higher food prices, owing to direct effects on crop cycles and growth. Climate change affects strong weather events, including summer storms, cyclones, and typhoons. It strengthens the El Niño and La Niña events, which are important factors in determining the direction of frequent seasonal storms and the intensification of inclement weather, such as heavy rain, in some countries. The Association of Southeast Asian Nations (ASEAN), including Thailand, is a global food producer that has experienced longer hot and dry spells in their areas of activity, resulting in water and food shortages. According to the Intergovernmental Panel on Climate Change (IPCC) report [1], risks due to climate change are increasing, and should be considered as a part of the 21st century climate system.

Climate change has affected Thailand in several ways; its impacts on the changes in temperature and rainfall are particularly notable. Thus, preparations to deal with and adapt to these changes are essential. In Thailand, when considering the change in temperature since the beginning of record keeping (1951) by the Thai Meteorological Department

(TMD) [2], it was found that the average temperature has been increasing. In 2021, the TMD [2] found that the annual average temperature was 27.5 °C, which was 0.4 °C higher than that in the previous 30-year period (1981–2010). Limsakul [3]'s research supports that day and night temperatures in Thailand have risen considerably and are likely to continue to increase. The northeastern region has shown a more rapid increase in temperature than other areas in Thailand. Increases in temperature impact agriculture, such as through water shortages and limiting plant growth. In addition, the northeastern region frequently experiences severe droughts. Because many people in the northeastern region are farmers, which comprise 46.6% of the population in Thailand [4], rises in temperature, especially during intense heatwaves, will significantly (and likely negatively) influence the means of living of many people in the region.

For the prediction of extreme temperature across Thailand, the statistical models employed by previous studies are rather simple and standard. Sharma and Babel [5] and Manomaiphiboon et al. [6] used regression models and the Mann-Kendall trend detection method. They reported that the extreme temperature tend to increase significantly for all stations examined in Thailand. Limsakul [3] used the empirical orthogonal function approach whereas Rodchuen et al. [7] used an ARMA model. The results of the aforementioned studies indicated that the frequency of hot (cool) temperature extremes has increased (decreased) and will continue in the near future.

Some studies, such as those by [5,8–10], have attempted to employ extreme value models to analyze extreme temperature data in Thailand. Seenoi et al. [10] modeled extreme temperatures in upper northeastern Thailand at nine meteorological stations using a generalized extreme value (GEV) distribution (GEVD). They estimated the parameters using the maximum likelihood estimation (MLE) method under stationary and non-stationary settings. None of the above-mentioned studies applied the model spatially, but only locally; here, local analysis means that the model was built independently for each weather station without considering the spatial dependency among nearby stations. It is generally known that spatial modeling of extreme values can reduce the mean squared error of prediction compared to local modeling [11–13]. The reason for the better performance of spatial modeling is that the spatial extreme model takes into account the spatial dependency among nearby stations, whereas the local extreme model is applied independently for each station without considering any events around nearby stations. However, spatial extreme modeling based on the max-stable process (MSP) requires intensive computation; thus, analysis with large number of locations is challenging [11]. Further, the MSP also requires a clear model specification. These disadvantages have prevented researchers from employing the MSP models to analyze extreme spatial data.

In our study, a spatial extreme value model based on an MSP was used and applied to the maximum temperature data in northeastern Thailand. The objective of the spatial analysis is to model a region where extreme events occur, and data are continuously stored [14]. Spatial analysis is an important in-depth analysis, as researchers sometimes cannot determine how independent or close each observation area is. Therefore, the spatial analysis was used to model spatial extreme events. This is a way to analyze spatial data and extreme value properties simultaneously. Yoon et al. [15] studied spatial GEV models within continuous local extreme events using observations of annual daily maximum rainfall data in northeastern Thailand, comprising 25 locations for the period 1982–2013. It was shown that the regional spatial GEV model reflects the spatial pattern well compared to the region-wide spatial GEV model as a local distribution of GEV and provides a strong return level. Several studies have suggested that MSPs are useful for statistical modeling of spatial extremes. These processes are natural extensions of multivariate extreme-value distributions to infinite dimensions [12,16–18]. Thus far, no study has been conducted on the spatial GEVD using observational temperature data in Thailand. Therefore, researcher are interested in studying it to prepare for problems that may arise from temperature changes.

Overall, in this study, we investigated the local GEV and spatial GEVD with extreme values of daily maximum temperature data in northeastern Thailand for 25 stations and predicted the return level of temperature in the 2, 10, 25, 50, and 100 year return periods. Section 2 presents the study areas and meteorology; Section 3 presents the modeling methodology, including the GEV model structure and spatial GEV model; Section 4 describes the results of both local GEV and spatial GEV models; and, finally, Section 5 presents the discussion and summarizes the conclusions of this study.

## 2. Study Area

The northeastern region of Thailand, which was considered as the study area, approximately 168,854 km$^2$, which is comparable to one-third of the total area of Thailand. This region is 120–400 m above sea level. The northeastern region is mostly a pan-like plateau divided into two large areas: the Korat River Basin, which includes the Mun and Chi rivers, and the Sakon Nakhon River Basin in the northern part of the region.

The general climate in Thailand has an average year-round temperature of 26–27 °C, with the lowest average temperature above 18 °C; therefore, Thailand is included in the tropical climate zone. The difference in climatic temperatures indicates that the months with the highest air temperature are April to early May, when the temperature reaches 40–41 °C, whereas the coldest months are December to January, when the temperature drops below 10 °C. The air temperature distribution was found to be high during April–September. When entering October, the air temperature gradually decreases and cools from November to February, and the weather warms again as March begins. From 1951 to 2012, the TMD [2] reported that the annual average temperature for the northeastern region was 28.6 °C, but the hottest air temperature was 43.9 °C in the Udon Thani Province in late April [19].

Figure 1 shows ridge line plots of the monthly average temperature from 1989 to 2019 for six stations (Nong Khai, Udon Thani, Sakon Nakhon, Khon Kaen, Chaiyaphum, and Nakhon Ratchasima) in the northeast region of Thailand, consistent with the previous description.

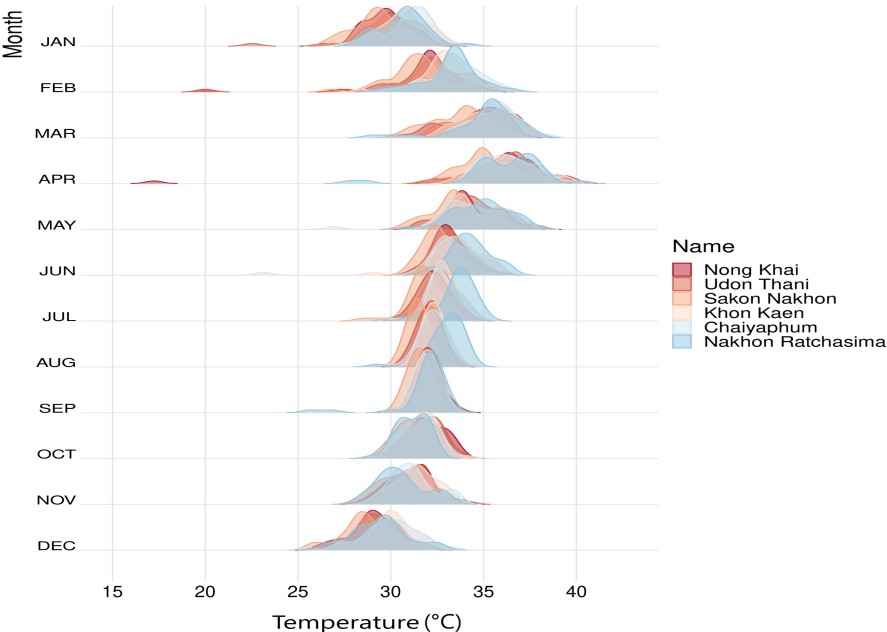

**Figure 1.** Ridge line plot of monthly average temperature data (unit: °C) from 1989 to 2019 for six stations in the northeast region of Thailand.

The northeastern region usually experiences a long period of warm weather because of its inland nature and tropical latitude zone. Between March and May, the peak temperature

is about 40 °C. In winter, monsoon winds from China cover Thailand, causing occasional cold weather and the temperature to drop to a relatively low value. Particularly in the northeast, the temperatures can drop to near zero °C [19].

In this study, we used yearly data to analyze GEV and then selected the annual maximum temperatures from the daily maximum temperature data obtained from the TMD [2]. The data records were for the duration of 1989 to 2019 from 25 stations in northeastern region of Thailand. The Table 1 shows the station names and identifications, including the altitude above sea level and descriptive statistics of the temperature data (unit: °C) for all 25 stations. All of the stations in the northeastern region of Thailand are shown in Figure 2. Although the weather stations were well distributed, some were located close to each other. Boxplots of the annual maximum temperature data for each station are plotted in Figure 3. This shows that some stations were located close to each other, but had different temperature distributions. For example, the Nakhon Ratchasima (431201), Pak Chong Agromet (431301), and Chok Chai (431401) stations are in the same province but have different temperature characteristics. This may be due to different topographies. The Pak Chong Agromet station is surrounded by mountains, and thus, the ambient temperature is relatively low. The Chok Chai Station is mostly in the highlands and has a river running through it. Therefore, both stations had lower temperatures than the Nakhon Ratchasima station.

**Table 1.** Temperature monitoring station and height above sea level for 25 stations and descriptive statistics of annual maximum temperature data (unit: °C) in northeastern region of Thailand. $N$ and $SD$ represent number of observations and standard deviation, respectively. $Q_1$ is 25th and $Q_3$ is 75th percentile.

| Station ID | Station Name | Latitude | Longitude | Altitude (m) | $N$ | Mean | $SD$ | Median | Min | Max | $Q_1$ | $Q_3$ |
|---|---|---|---|---|---|---|---|---|---|---|---|---|
| 352201 | Nong Khai | 17°52′ | 102°43′ | 167 | 31 | 40.9 | 1.20 | 40.6 | 38.9 | 43.3 | 40.0 | 41.9 |
| 353201 | Loei | 17°27′ | 101°44′ | 246 | 31 | 40.7 | 1.38 | 40.4 | 38.6 | 43.4 | 39.8 | 41.7 |
| 353301 | Loei Agromet | 17°24′ | 101°43′ | 311 | 31 | 39.2 | 7.40 | 40.5 | 38.0 | 43.5 | 39.6 | 41.2 |
| 354201 | Udonthani | 17°23′ | 102°43′ | 177 | 31 | 40.9 | 1.18 | 41.0 | 38.3 | 43.0 | 40.1 | 41.8 |
| 356201 | Sakon Nakhon | 17°09′ | 104°08′ | 168 | 31 | 40.0 | 1.08 | 40.0 | 38.0 | 41.7 | 39.1 | 41.0 |
| 356301 | Sakon Nakhon Agromet | 17°07′ | 104°03′ | 238 | 31 | 38.8 | 7.29 | 40.0 | 37.9 | 42.5 | 39.3 | 41.0 |
| 357201 | Nakhon Phanom | 17°25′ | 104°47′ | 141 | 31 | 39.5 | 1.27 | 39.2 | 37.5 | 42.1 | 38.5 | 40.5 |
| 357301 | Nakhon Phanom Agromet | 16°26′ | 104°47′ | 142 | 31 | 38.4 | 7.23 | 39.8 | 37.3 | 42.1 | 38.8 | 40.6 |
| 381201 | Khon Kaen | 16°27′ | 102°49′ | 168 | 31 | 40.6 | 0.98 | 40.8 | 38.5 | 42.4 | 39.8 | 41.3 |
| 381301 | Tahpra Agromet | 16°20′ | 104°43′ | 171 | 31 | 39.4 | 7.39 | 40.6 | 38.3 | 42.7 | 40.0 | 41.3 |
| 383201 | Mukdaharn | 16°32′ | 104°43′ | 162 | 31 | 40.7 | 1.02 | 40.8 | 38.9 | 42.5 | 40.0 | 41.6 |
| 387401 | Maha Sarakham | 16°14′ | 103°04′ | 161 | 31 | 40.8 | 1.03 | 40.6 | 39.0 | 43.3 | 40.2 | 41.6 |
| 403201 | Chaiyaphum | 15°48′ | 102°02′ | 209 | 31 | 40.4 | 1.12 | 40.5 | 38.1 | 42.6 | 39.5 | 41.0 |
| 405201 | Roiet | 16°03′ | 103°41′ | 147 | 31 | 39.8 | 1.06 | 39.7 | 38.0 | 42.3 | 39.0 | 40.4 |
| 405301 | Roiet Agromet | 16°04′ | 103°37′ | 161 | 31 | 38.4 | 7.21 | 39.8 | 35.9 | 41.2 | 39.0 | 40.3 |
| 407301 | Ubon Ratchatani Agromet | 15°23′ | 105°03′ | 118 | 31 | 38.8 | 7.29 | 40.1 | 38.0 | 42.4 | 39.2 | 40.7 |
| 407501 | Ubon Ratchatani | 15°15′ | 104°52′ | 126 | 31 | 40.2 | 1.15 | 40.3 | 37.9 | 42.6 | 39.2 | 41.0 |
| 409301 | Si Sa Ket | 15°02′ | 104°15′ | 134 | 31 | 38.8 | 7.27 | 40.0 | 38.4 | 42.5 | 39.7 | 40.6 |
| 431201 | Nakhon Ratchasima | 14°57′ | 102°04′ | 204 | 31 | 40.4 | 1.21 | 40.6 | 37.9 | 43.2 | 39.6 | 41.3 |
| 431301 | Pak Chong Agromet | 14°38′ | 101°19′ | 551 | 31 | 36.2 | 6.81 | 37.5 | 35.0 | 39.4 | 36.5 | 38.2 |
| 431401 | Chok Chai | 14°43′ | 102°10′ | 187 | 31 | 39.6 | 0.95 | 39.5 | 38.1 | 42.5 | 39.0 | 40.0 |
| 432201 | Surin | 14°53′ | 103°30′ | 147 | 31 | 39.4 | 0.97 | 39.3 | 38.0 | 42.0 | 38.7 | 39.8 |
| 432301 | Surin Agromet | 14°53′ | 103°27′ | 144 | 31 | 39.0 | 7.32 | 40.2 | 38.0 | 43.3 | 39.5 | 41.0 |
| 432401 | Tha Tum | 15°19′ | 103°41′ | 144 | 31 | 40.3 | 1.16 | 40.2 | 38.6 | 42.3 | 39.2 | 41.3 |
| 436401 | Nang Rong | 14°35′ | 102°48′ | 185 | 31 | 40.4 | 1.03 | 40.4 | 38.2 | 43.0 | 39.8 | 41.1 |

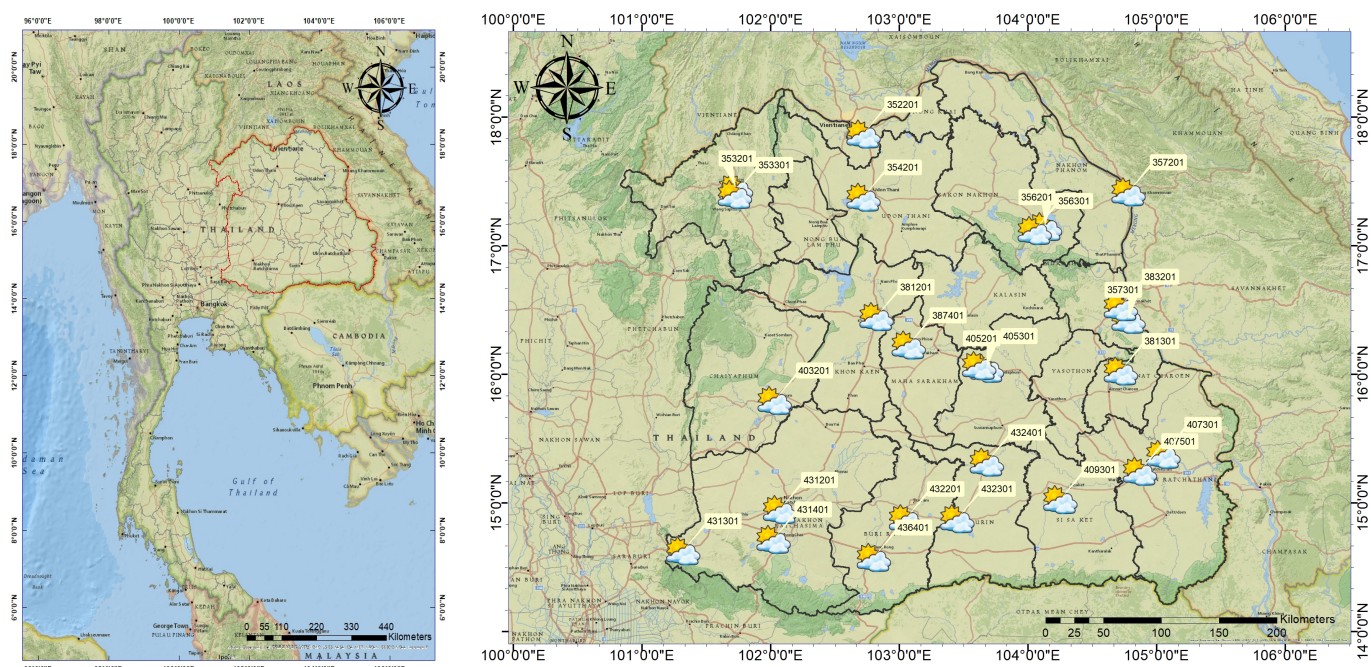

**Figure 2.** Locations of 25 observation weather stations in northeast region of Thailand with labels above each location showing the ID of each station.

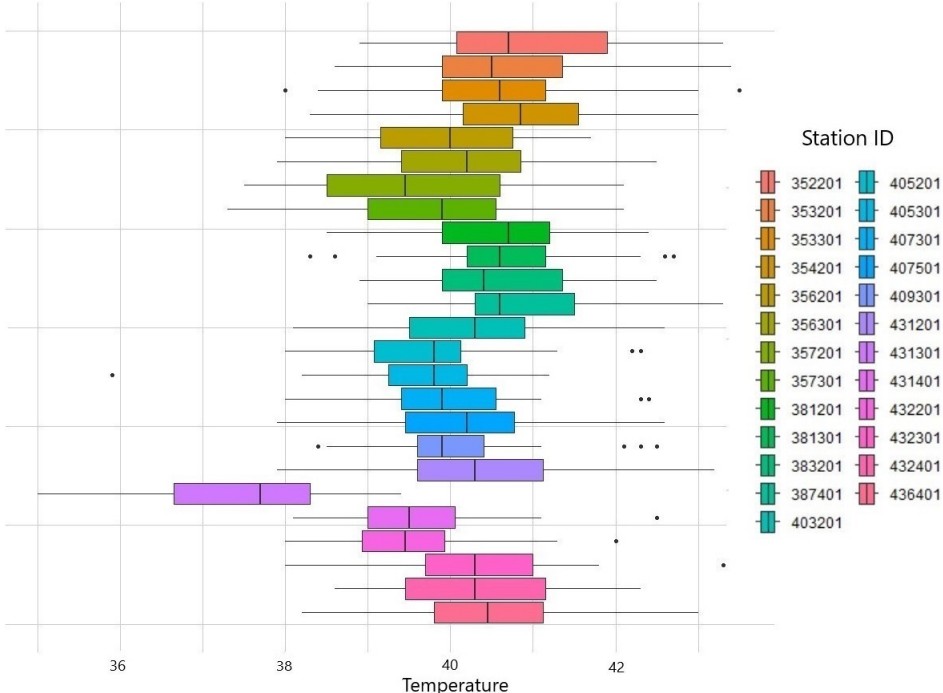

**Figure 3.** Boxplots of annual maximum temperature data for each station in northeastern region.

## 3. Methodology

Analysis of the extreme value model with extreme value theory (EVT) can be divided into two types according to the nature of the selection of the extreme value data: GEVD and generalized Pareto distribution (GPD). Modeling with GEVD is suitable for analyzing extreme values over time periods of interest, such as daily, weekly, monthly, quarterly, and yearly. GPD analysis is suitable when there is a large amount of daily data. In this study,

we analyzed the GEVD by selecting yearly maximum values from daily temperature data recorded from 1989 to 2019 for 25 stations.

### 3.1. GEVD

The GEVD developed by Jenkinson [20] is flexible to three other distributions: Gumbel, Fréchet, and Weibull distributions. Assuming that $X_i$, where $i = 1, 2, \cdots, n$ are independent random variables and have the same probability density function $F(x)$, the maximum value of the random variable is $X_{(n)} = Max(X_1, X_2, \cdots, X_n)$.

### 3.1.1. Local GEVD

The cumulative distribution function (CDF) of the GEVD is as follows [11]:

$$F(x) = exp\left(-\left(1 + \xi\frac{x - \mu}{\sigma}\right)^{-1/\xi}\right), \ 1 + \xi\frac{x - \mu}{\sigma} > 0 \tag{1}$$

where $\mu, \sigma$, and $\xi$ denote the location, scale, and shape parameters, respectively. The case with $\xi \to 0$ is the Gumbel distribution

$$F(x) = exp\left(-\left(1 + \xi\frac{x - \mu}{\sigma}\right)\right), \ -\infty < x < \infty \tag{2}$$

The cases with $\xi > 0$ and $\xi < 0$ are known as Fréchet and negative Weibull distributions, respectively.

### 3.1.2. MLE

We estimated the parameters using the MLE method. Assuming that $X_1, X_2, \cdots,$ and, $X_n$ are independent variables and have a GEVD, then the log-likelihood function can be written as follows:

$$l(\mu, \sigma, \xi) = -mlog\sigma - \left(1 + \frac{1}{\xi}\right)\sum_{i=1}^{n} log\left[1 + \xi\left(\frac{x_i - \mu}{\sigma}\right)\right] - \sum_{i=1}^{n}\left[1 + \xi\left(\frac{x_i - \mu}{\sigma}\right)\right]^{-\frac{1}{\xi}}, \tag{3}$$

where $\xi \neq 0$ and $1 + \xi\left(\frac{x_i - \mu}{\sigma}\right) > 0$ for $i = 1, \cdots, n$. For the case $\xi = 0$, it is the limit of the Gumbel distribution; then, the log-likelihood function becomes

$$l(\mu, \sigma) = -mlog\sigma - \sum_{i=1}^{n}\left(\frac{x_i - \mu}{\sigma}\right) - \sum_{i=1}^{n}exp\left\{-\left(\frac{x_i - \mu}{\sigma}\right)\right\}. \tag{4}$$

In practice, it is easier to maximize the log-likelihood function. We obtained the values of $\hat{\mu}$, $\hat{\sigma}$, and $\hat{\xi}$ from Equations (3) and (4).

The goodness-of-fit test used for this study was the Kolmogorov-Smirnov (KS) test, which is obtained by transforming to a standard Gumbel distribution, defined by [11].

$$\tilde{x} = \frac{1}{\hat{\xi}} \log\left\{1 + \hat{\xi}\left(\frac{x - \hat{\mu}}{\hat{\sigma}}\right)\right\}, \tag{5}$$

with probability distribution function $Pr\{\tilde{x} \leq x\} = \exp\{-e^{-x}\}, x \in \mathbb{R}$.

### 3.1.3. Return Level Estimation

After the parameters were estimated using the maximum likelihood method, the return level value $z_p$ for $0 < p < 1$, where $z_p$ is defined as the value expected to exceed the average once every $1/p$ Coles [11], can be calculated as follows:

$$
\hat{z}_p \begin{cases} \hat{\mu} - \dfrac{\hat{\sigma}}{\hat{\xi}}\left[1 - y_p^{-\hat{\xi}}\right], & \text{for } \hat{\xi} \neq 0, \\ \hat{\mu} - \hat{\sigma} log y_p, & \text{for } \hat{\xi} = 0, \end{cases} \tag{6}
$$

where $y_p = -log(1 - p)$. The standard error of the return level was calculated using the delta method. The extreme value model has a preliminary agreement that the distribution of the GEV data at each station is independent of one another; thus, our data are a multivariate series, since data from several locations were recorded. Therefore, the extreme values theory approaches are insufficient and were, thus, modeled using the spatial extreme value, as presented [21].

### 3.2. Spatial GEVD

In this study, we created a spatial model for extreme values via an MSP using Schlater's characterization. The MSP is an infinite-dimensional generalization of multivariate EVT distribution. The concept of spatial dependence is that everything is interrelated, but the nearer ones are more closely related than the distant ones [21], as shown in Figure 4.

For data analysis, we used linear transformations to provide geographic information of the station, as in Equation (7).

$$
x_{new} = \frac{[x - min(x)]}{[max(x) - min(x)]} \tag{7}
$$

to make the inverse calculation of the Hessian matrix more stable [15].

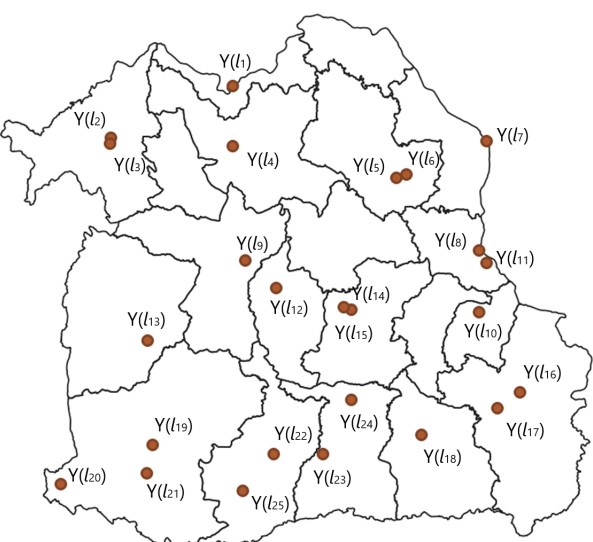

**Figure 4.** Spatial location in northeast region of Thailand.

Let $M(l,t)$ be the daily maximum temperature data for the location ($l$) and period ($t$) in the spatial domain $D \subset \mathbb{R}^2$ [21]. Then, the distribution of $M(l,t)$ is:

$$
M(l,t) = GEV(\mu(l,t), \sigma(l,t), \xi(l,t)), \tag{8}
$$

where $\mu(l,t), \sigma(l,t),$ and $\xi(l,t)$ are the locations, scale, and shape parameters from the GEVD. A stochastic process $Z(r)$ is defined as an MSP if successions of continuous function $a_n(r)$ and $b_n(r)$ exist, as in Equation (9).

$$Z(r) = \frac{\max\limits_{i=1,\cdots,n} Z_i(r) - b_n(r)}{a_n(r)}, r \in \mathbb{R} \tag{9}$$

In the present study, we used Schlater's method to rewrite Equation (9) as Equation (10):

$$Z(x) = \max\limits_{i \geq 1}(U_i Y_i(x)), x \in X, \tag{10}$$

where $U_i$ are the points of a Poisson process at $(0, +\infty)$, and $Y_1(x), \cdots, Y_n(x)$ are independent replications of the stochastic process $Y(x)$. Portero Serrano et al. [22] presented at bivariate CDF, as follows:

$$P(z_i, z_j) = exp\left\{ -\frac{1}{2}\left(\frac{1}{z_i} + \frac{1}{z_j}\right)\left(1 + \sqrt{1 - 2(\rho(h) + 1)\frac{z_i z_j}{(z_i + z_j)^2}}\right) \right\}, \tag{11}$$

where $\rho(h)$ is the correlation function and the distance $h$ separates the different maxima:

$$\rho(h) = v exp\left( -\left(\frac{h}{\tau}\right)^\eta \right), 0 < \eta \leq 2, \tau > 0, 0 \leq v \leq 1, h > 0, \tag{12}$$

where $\tau$ and $\eta$ are the scale and shape parameters. The parameter $v$ represents the nugget effect of measurement errors and microscale variations in the data [13]. The correlation function $\rho(h)$ between two random $(X, Y)$ items should have a positive correlation for $\rho = 1$ as a full dependency, with this function being constrained by the extremal coefficient $(\theta)$ describing the characteristics of the matrix of the dependencies tail and the extremal coefficient function $\theta(h) = 1 + [\{1 - \rho(h)\}/2]^{1/2}$ [21–24].

To select the best trend surface model, we used the Takeuchi information criterion (TIC) [25] and, chose the smallest available one. Through a model selection procedure, we built a regression-based form for the location, scale, and shape parameters, as follows:

$$\begin{aligned}
\hat{\mu}(x) &= \mu_0 + \mu_{lon}lon(x) + \mu_{lon}lon(x)^2 + \mu_{lat}lat(x) + \mu_{lat}lat(x)^2 + \mu_{alt}alt(x) \\
\hat{\sigma}(x) &= \sigma_0 + \sigma_{lat}lat(x) + \sigma_{alt}alt(x) \\
\hat{\xi}(x) &= \xi_0
\end{aligned} \tag{13}$$

where these models are several combinations of models with the latitude, longitude, and altitude of the station at which the data were observed $x$. The shape parameter $\xi(x)$ was treated as a constant. The parameters $\psi$ of the model were identified by the pairwise marginal density and estimated by maximizing a composite log-likelihood function, as follows [24]:

$$l_p(\hat{\psi}) = \sum_{i=1}^{n} \sum_{\{j<k:y_j,y_k \in Y_i\}} logf(y_i, y_k; \psi), \tag{14}$$

and TIC function is given by

$$TIC = -2l_p(\hat{\psi}) + 2tr\left\{ H(\hat{\psi})^{-1} J(\hat{\psi}) \right\}, \tag{15}$$

where

$$H(\hat{\psi}) = -\frac{\partial^2 logf(z_{ik}, z_{jk}; \hat{\psi})}{\partial \psi \partial \psi^T}$$

$$J(\hat{\psi}) = -\sum_{k=1}^{K} \frac{\partial logf(z_{ik}, z_{jk}; \hat{\psi})}{\partial \psi} \frac{\partial logf(z_{ik}, z_{jk}; \psi)}{\partial \psi^T},$$

where $i = 1, 2, \cdots, n, j = 1, 2, \cdots, m$ and the return level analysis concept are used to estimate the future extreme temperature.

## 4. Results

### *4.1. GEVD*

From Table 2, the 95% confidence interval estimated the contour shape parameters from the local GEV of the study area, indicating the optimal distribution at each station. There are 10 stations with the contour parameter, and positive coverage means that the tail is skewed to the right; therefore, the optimal distribution is the Gumbel distribution. There are 15 stations in which the shape parameter approximation is in the zero range, that is, the Weibull distribution. We performed our model with the KS test statistic at a significance level of 0.05 by comparing the CDF values of the sample data with the CDF values of the actual data. It was found that the Gumbel and Weibull distributions were appropriate for the data.

**Table 2.** Point parameter estimates in 95 confidence interval, appropriate distribution and p-value of Kolmogorov–Smirnov (KS) test of some stations from local GEV.

| Station | Parameter Estimate | | | Distribution | *p*-Value of KS |
|---|---|---|---|---|---|
| | $\hat{\mu}$(s.e) | $\hat{\sigma}$(s.e) | $\hat{\xi}$(s.e) | | |
| | CI 95% | CI 95% | CI 95% | | |
| 353201 | 40.19 (0.24) (39.72, 40.66) | 1.15 (0.18) (0.79, 1.51) | −0.33 (0.18) (−0.69, 0.03) | Gumbel | 0.96 |
| 353301 | 40.12 (0.23) (39.67, 40.57) | 1.12 (0.17) (0.78, 1.45) | −0.31 (0.16) (−0.63, 0.00) | Gumbel | 0.91 |
| 354201 | 40.51 (0.21) (40.09, 40.93) | 1.08 (0.15) (0.79, 1.38) | −0.35 (0.11) (−0.57, −0.13) | Weibull | 0.99 |
| 356201 | 39.81 (0.24) (39.35, 40.27) | 1.18 (0.23) (0.73, 1.64) | −0.77 (0.20) (−1.16, −0.38) | Weibull | 0.86 |
| 357201 | 39.04 (0.24) (38.57, 39.52) | 1.16 (0.18) (0.79, 1.52) | −0.23 (0.18) (−0.58, 0.12) | Gumbel | 0.85 |
| 381301 | 40.51 (0.17) (40.17, 40.84) | 0.85 (0.13) (0.60, 1.09) | −0.29 (0.14) (−0.57, −0.02) | Weibull | 0.97 |
| 405201 | 39.83 (0.21) (39.43, 40.23) | 1.01 (0.15) (0.72, 1.31) | −0.31 (0.14) (−0.59, −0.03) | Weibull | 0.99 |
| 407501 | 39.87 (0.19) (39.49, 40.24) | 0.94 (0.14) (0.66, 1.23) | −0.34 (0.15) (−0.63, −0.04) | Weibull | 0.99 |
| 431201 | 40.02 (0.21) (39.61, 40.43) | 1.08 (0.15) (0.78, 1.38) | −0.45 (0.10) (−0.64, −0.25) | Weibull | 0.98 |
| 431301 | 38.47 (0.19) (38.09, 38.84) | 0.97 (0.14) (0.70, 1.24) | −0.38 (0.12) (−0.61, −0.14) | Weibull | 0.89 |
| 432201 | 39.05 (0.15) (38.75, 39.36) | 0.76 (0.11) (0.55, 0.98) | −0.18 (0.13) (−0.44, 0.08) | Gumbel | 0.92 |

The return level estimates of the study areas from the local GEVD calculated from Equation (6) are shown in Table 3, indicating that Nong Khai Station (352201) had higher return levels for temperature than the other stations, whereas Surin Station (432201) had the lowest return levels for every return period.

**Table 3.** Return levels (unit: °C) for several return periods ($T = 5, 10, 25, 50,$ and $100$) with standard error in parentheses for each station from local generalized extreme value distribution.

| Station | 5 Years | 10 Years | 25 Years | 50 Years | 100 Years |
|---|---|---|---|---|---|
| 352201 | 41.87 (0.07) | 42.41 (0.59) | 42.99 (0.19) | 43.35 (0.35) | 43.66 (0.59) |
| 353201 | 41.55 (0.06) | 42.01 (0.29) | 42.46 (0.11) | 42.71 (0.18) | 42.91 (0.29) |
| 353301 | 41.46 (0.05) | 41.92 (0.26) | 42.38 (0.10) | 42.64 (0.17) | 42.84 (0.26) |
| 354201 | 41.77 (0.04) | 42.19 (0.10) | 42.59 (0.05) | 42.81 (0.07) | 42.98 (0.10) |
| 356201 | 40.86 (0.02) | 41.07 (0.004) | 41.21 (0.004) | 41.27 (0.003) | 41.30 (0.004) |
| 356301 | 40.84 (0.02) | 41.12 (0.02) | 41.33 (0.01) | 41.43 (0.01) | 41.49 (0.02) |
| 357201 | 40.51 (0.08) | 41.08 (0.63) | 41.67 (0.21) | 42.03 (0.38) | 42.34 (0.63) |
| 357301 | 40.53 (0.07) | 41.08 (0.52) | 41.65 (0.18) | 42.00 (0.32) | 42.29 (0.52) |
| 381201 | 41.19 (0.02) | 41.45 (0.06) | 41.66 (0.03) | 41.76 (0.04) | 41.83 (0.06) |
| 381301 | 41.44 (0.03) | 41.71 (0.14) | 41.95 (0.06) | 42.07 (0.09) | 42.16 (0.14) |
| 383201 | 41.54 (0.02) | 41.90 (0.02) | 42.27 (0.01) | 42.48 (0.02) | 42.65 (0.02) |
| 387401 | 41.48 (0.02) | 41.75 (0.06) | 41.98 (0.02) | 42.09 (0.04) | 42.17 (0.06) |
| 403201 | 41.04 (0.04) | 41.47 (0.17) | 41.88 (0.07) | 42.11 (0.11) | 42.30 (0.17) |
| 405201 | 40.99 (0.04) | 41.35 (0.17) | 41.68 (0.07) | 41.85 (0.12) | 41.99 (0.17) |
| 405301 | 40.97 (0.03) | 41.35 (0.07) | 41.73 (0.04) | 41.94 (0.05) | 42.11 (0.07) |
| 407301 | 41.03 (0.05) | 41.49 (0.30) | 41.97 (0.11) | 42.26 (0.19) | 42.51 (0.30) |
| 407501 | 40.98 (0.04) | 41.36 (0.14) | 41.72 (0.06) | 41.92 (0.09) | 42.08 (0.14) |
| 409301 | 40.93 (0.05) | 41.40 (0.22) | 41.89 (0.10) | 42.20 (0.15) | 42.46 (0.22) |
| 431201 | 40.98 (0.03) | 41.36 (0.03) | 41.72 (0.03) | 41.92 (0.03) | 42.08 (0.03) |
| 431301 | 40.93 (0.03) | 41.40 (0.08) | 41.89 (0.04) | 42.20 (0.05) | 42.46 (0.08) |
| 431401 | 40.22 (0.03) | 41.29 (0.23) | 41.92 (0.08) | 42.30 (0.14) | 42.46 (0.23) |
| 432201 | 39.58 (0.04) | 39.94 (0.25) | 40.28 (0.09) | 40.46 (0.16) | 40.60 (0.25) |
| 432301 | 40.19 (0.03) | 40.56 (0.05) | 40.94 (0.02) | 41.18 (0.03) | 41.39 (0.05) |
| 432401 | 40.06 (0.07) | 40.47 (0.50) | 40.91 (0.17) | 41.20 (0.30) | 41.44 (0.50) |
| 436401 | 41.00 (0.02) | 41.31 (0.02) | 41.59 (0.01) | 41.72 (0.01) | 41.82 (0.02) |

### 4.2. Spatial GEVD

We modeled the spatial model for extreme temperature data for 25 stations in northeast Thailand via an MSP using Schalater's method with a powered exponential covariance function $\rho(h)$ [15,23]. Table 4 shows the parameter estimation with standard error in parenthesis which was calculated via the MSP method referenced in Equation (14). Through a model selection from minimum TIC criterion among possible forms of $\mu(x)$ and $\sigma(x)$ from Equation (15), we obtained the location and scale parameters as in Table 4. From this table, $lat^2$ represents the squared term of the latitude of $x$ and the shape parameter is treated as a constant.

**Table 4.** The parameter estimation with standard error in parenthesis obtained from spatial GEV models. Calculated via the max-stable process method referenced in Equation (14).

| | | | Parameter Estimate |
|---|---|---|---|
| $\rho(h)$ | | $\nu$ | 0.0289 (0.0076) |
| | | $\tau$ | 0.2083 (0.0497) |
| $\mu(x)$ | | $\mu_0$ | 41.8774 (0.2481) |
| | | $lat$ | 0.6843 (0.1084) |
| | | $lat^2$ | −1.5736 (0.1749) |
| | | $alt$ | −0.0096 (0.0008) |
| $\sigma(x)$ | | $\sigma_0$ | 1.0924 (0.1101) |
| | | $lat$ | −0.0002 (0.0004) |
| $\xi$ | | $\xi$ | −0.2667 (0.0415) |

Figure 5 presents the scatter plots of $\mu$, $\sigma$, and $\xi$ from the GEVD and MSP for 25 stations, independently corresponding to the model selection procedure. Based on the selected model, we obtained estimates of location ($\mu$), scale ($\sigma$), and shape ($\xi$) parameters for regional spatial GEV in Table 5.

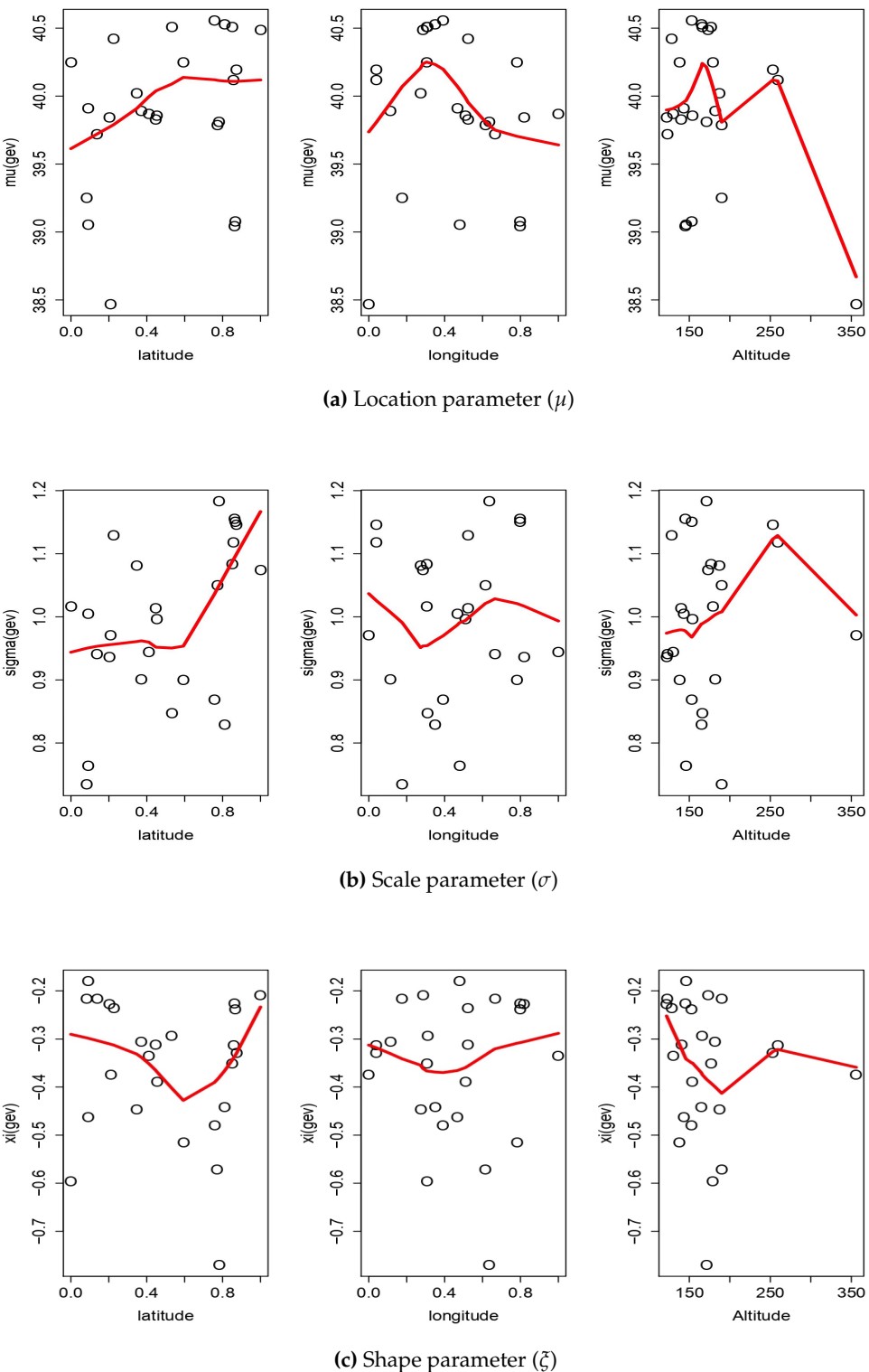

**Figure 5.** Relationships between geographic covariance and GEV parameters where parameter estimates were obtained from GEV models for 25 stations independently.

**Table 5.** Estimated parameters and standard error values (in parenthesis) obtained from regional spatial GEV model.

| Station | Location $\hat{\mu}(x)$ | Scale $\hat{\sigma}(x)$ | Shape $\hat{\xi}(x)$ |
|---|---|---|---|
| 352201 | 40.446 (0.15) | 1.062 (0.07) | −0.267 (0.04) |
| 353201 | 39.974 (0.15) | 1.047 (0.08) | −0.267 (0.04) |
| 353301 | 39.906 (0.15) | 1.046 (0.08) | −0.267 (0.04) |
| 354201 | 40.274 (0.14) | 1.061 (0.07) | −0.267 (0.04) |
| 356201 | 39.765 (0.13) | 1.062 (0.07) | −0.267 (0.04) |
| 356301 | 39.607 (0.13) | 1.059 (0.07) | −0.267 (0.04) |
| 357201 | 39.817 (0.14) | 1.067 (0.07) | −0.267 (0.04) |
| 357301 | 39.742 (0.14) | 1.065 (0.07) | −0.267 (0.04) |
| 381201 | 40.292 (0.14) | 1.063 (0.07) | −0.267 (0.04) |
| 381301 | 40.154 (0.14) | 1.063 (0.07) | −0.267 (0.04) |
| 383201 | 39.723 (0.13) | 1.068 (0.07) | −0.267 (0.04) |
| 387401 | 40.304 (0.14) | 1.065 (0.07) | −0.267 (0.04) |
| 403201 | 40.198 (0.15) | 1.060 (0.07) | −0.267 (0.04) |
| 405201 | 40.010 (0.13) | 1.067 (0.07) | −0.267 (0.04) |
| 405301 | 39.903 (0.13) | 1.065 (0.07) | −0.267 (0.04) |
| 407301 | 39.332 (0.14) | 1.069 (0.08) | −0.267 (0.04) |
| 407501 | 39.553 (0.14) | 1.071 (0.08) | −0.267 (0.04) |
| 409301 | 39.739 (0.13) | 1.070 (0.08) | −0.267 (0.04) |
| 431201 | 39.883 (0.13) | 1.059 (0.08) | −0.267 (0.04) |
| 431301 | 38.590 (0.17) | 1.029 (0.10) | −0.267 (0.04) |
| 431401 | 39.825 (0.14) | 1.059 (0.07) | −0.267 (0.04) |
| 432201 | 39.779 (0.14) | 1.066 (0.07) | −0.267 (0.04) |
| 432301 | 39.828 (0.14) | 1.067 (0.07) | −0.267 (0.04) |
| 432401 | 39.974 (0.13) | 1.070 (0.08) | −0.267 (0.04) |
| 436401 | 39.672 (0.14) | 1.060 (0.07) | −0.267 (0.04) |

Table 6 and Figure 6 show the return levels corresponding to 2, 25, 50, and 100 year return periods, which were obtained from an MSP using the regional models for 25 stations in the northeast region of Thailand. Table 6 shows that the stations with the highest return levels were Nong Khai (352201), Maha Sarakham (387401) and Khon Kaen (381201), while Pak Chong Agromet station (431301) had the lowest return temperature levels. From Figure 6, we can see that the MSP model collects geographical and covariate information well across the region. We used a Kriging and inverse distance weighting (IDW) technique for interpolation in drawing Figure 6. Then we compared the IDW and Kriging techniques and found that the IDW method was more effective. See detailed results in supplementary material. This result is consistent with the TMD report that these stations are relatively flat and the climate is classified as tropical [2].

**Table 6.** Return levels (unit: °C) for several return periods (T = 2, 10, 25, 50, and 100) for each station from spatial generalized extreme value distribution.

| Station | 5 Years | 10 Years | 25 Years | 50 Years | 100 Years |
|---|---|---|---|---|---|
| 352201 | 40.82 | 42.24 | 42.73 | 43.02 | 43.26 |
| 353201 | 40.34 | 41.75 | 42.23 | 42.51 | 42.75 |
| 353301 | 40.27 | 41.68 | 42.16 | 42.44 | 42.68 |
| 354201 | 40.64 | 42.07 | 42.56 | 42.85 | 43.09 |
| 356201 | 40.14 | 41.56 | 42.05 | 42.34 | 42.58 |
| 356301 | 39.98 | 41.40 | 41.88 | 42.17 | 42.41 |

**Table 6.** *Cont.*

| Station | 5 Years | 10 Years | 25 Years | 50 Years | 100 Years |
|---------|---------|----------|----------|----------|-----------|
| 357201 | 40.19 | 41.62 | 42.11 | 42.40 | 42.64 |
| 357301 | 40.11 | 41.54 | 42.03 | 42.33 | 42.56 |
| 381201 | 40.66 | 42.09 | 42.58 | 42.87 | 43.11 |
| 381301 | 40.52 | 41.95 | 42.44 | 42.73 | 42.97 |
| 383201 | 40.10 | 41.53 | 42.02 | 42.31 | 42.55 |
| 387401 | 40.68 | 42.11 | 42.60 | 42.89 | 43.13 |
| 403201 | 40.57 | 41.99 | 42.48 | 42.77 | 43.01 |
| 405201 | 40.38 | 41.82 | 42.31 | 42.60 | 42.84 |
| 405301 | 40.27 | 41.70 | 42.19 | 42.49 | 42.72 |
| 407301 | 39.71 | 41.14 | 41.63 | 41.92 | 42.17 |
| 407501 | 39.93 | 41.36 | 41.86 | 42.15 | 42.39 |
| 409301 | 40.11 | 41.55 | 42.04 | 42.34 | 42.58 |
| 431201 | 40.25 | 41.68 | 42.16 | 42.45 | 42.69 |
| 431301 | 38.95 | 40.33 | 40.80 | 41.09 | 41.32 |
| 431401 | 40.20 | 41.62 | 42.10 | 42.39 | 42.63 |
| 432201 | 40.15 | 41.58 | 42.07 | 42.37 | 42.61 |
| 432301 | 40.20 | 41.63 | 42.12 | 42.41 | 42.65 |
| 432401 | 40.35 | 41.78 | 42.28 | 42.57 | 42.81 |
| 436401 | 40.04 | 41.47 | 41.95 | 42.24 | 42.48 |

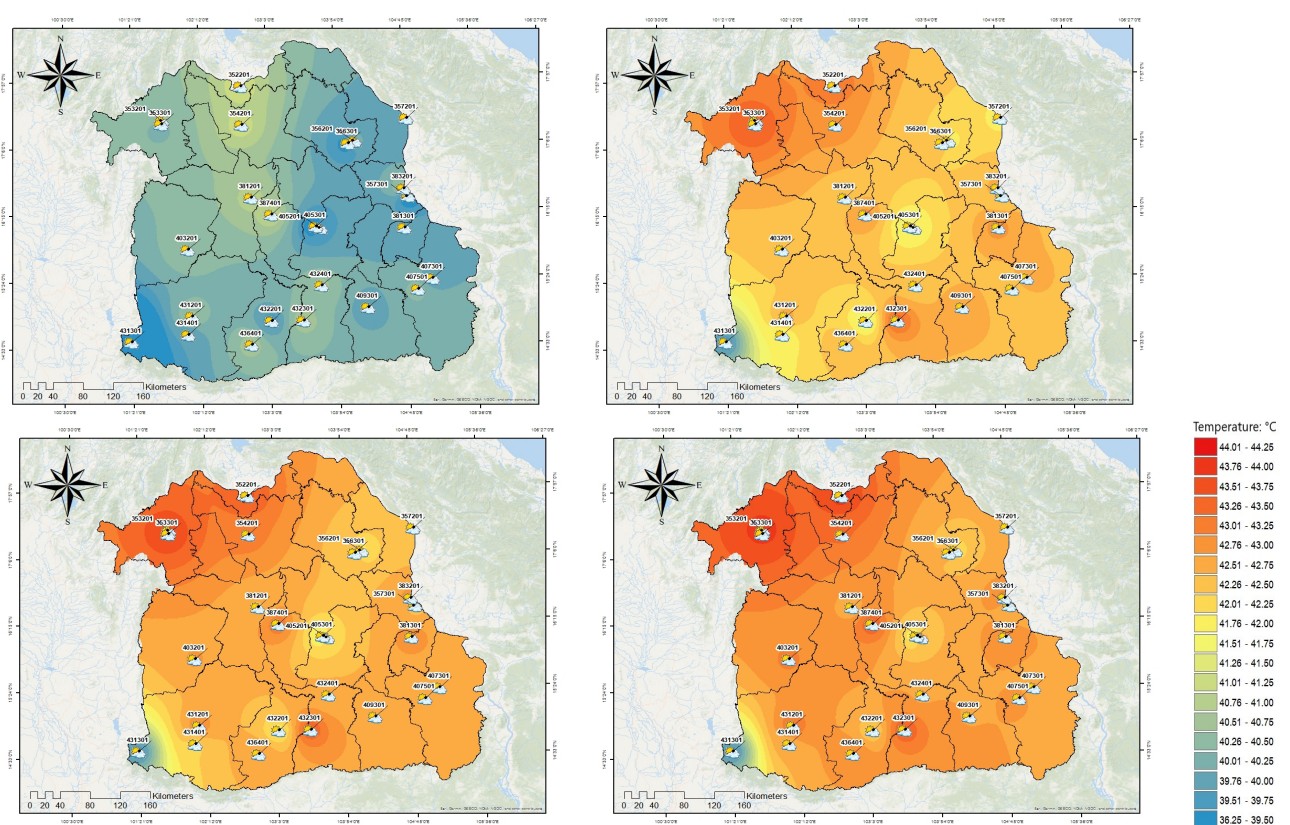

**Figure 6.** Return level maps for 25 stations corresponding to 2, 25, 50, and 100 year return periods in northeast region of Thailand (unit: °C). Return levels were obtained from max-stable process models.

## 5. Discussion and Conclusions

In this study, we used GEV and spatial GEV modeling using the MSP to analyze the annual maximum temperatures in northeastern Thailand at 25 stations. We used Schlathers

method with a powered exponential correlation function to build a regional model. In addition, we predicted the return levels of the observation data for several return periods using regional models and drew return level maps. This study showed that 15 stations had Weibull distributions and 10 had Gumbel distributions, which were suitable for the significant KS tests for all stations for the local GEV model. The return level estimates of the study areas from the local GEV showed that Nong Khai (352201) had higher return levels than the other stations for every return period. For the spatial GEV model, we analyzed extreme data using MSP with latitude (lat), longitude (lon), and height above sea level (altitude, alt) variables. The equations suitable for modeling are as follows:

$$\hat{\mu}(x) = \mu_0 + \mu_{lat} lat(x) + \mu_{lat} lat(x)^2 + \mu_{alt} alt(x)$$
$$\hat{\sigma}(x) = \sigma_0 + \sigma_{lat} lat(x)$$

We set the shape parameter as constant for the spatial GEV models. Thus, future research could consider a nonconstant shape parameter $\xi(x)$, which may require more samples. Building the best regression-based form for the location and scale parameters requires a substantial computational burden. We selected covariates at the minimum TIC from many possible models. We employed a stage-wise selection procedure to select the best regression-based form from all the possible combinations. However, we observed failure of the TIC computation in a few cases, which made the procedure unstable. In this regard, future research should develop a fast and stable algorithm for the better selection of variables. The Figure 6 shows the MSP model of the spatial GEV, which collects geographical and covariate information across the region. The stations with the highest return temperatures were Nong Khai (352201), Maha Sarakham (387401), and Khon Kaen (381201), whereas Pak Chong Agromet (431301) had the lowest return levels in every return period. The estimation results of the return level of our study is consistent with Limsakul's [3] research showing that Thailand's temperature tends to increase, as well as Seenoi's [10] research supporting that temperatures in upper northeastern Thailand tend to increase.

Due to the continually increasing estimate of the return level, the northeastern region of the country may be affected by some risky extreme events. These are, for example, severe droughts and intense heatwaves that could result in forest fires or significantly fewer agricultural products. This is coupled with the possibility of overall temperatures exceeding 35 °C and the spread of pests and plant pathogens.

The approach of this study can be applied to the field of numerical model outputs of climate systems and compared to a model fitted to climate observations. Future studies should seek a significant covariate that affects temperature. The latitude, longitude, and altitude data are considered important; however, we may have missed more important covariates for extreme temperatures in this study, such as topographic aspects and coastal proximity. One of the reasons why the global spatial GEV model is not a good reflection is that the geographic covariates do not reflect the characteristics of the region. If the covariates that have influence are considered in the model, it is expected that the explanatory power of the model will be further enhanced. Furthermore, the detection and prediction of changes in climate, including trends in extreme temperatures [13] and extreme wind speeds [26], will be performed. In addition, the MSP modeling approach for different time periods can be used to determine climate change extremes in Thailand.

Modeling the entire distribution (including the maximum, mean, and minimum simultaneously) of a climate variable for a spatial field and using it to detect changes (e.g., shifts in means and standard deviations) may be more reliable and valuable than modeling only extremes [27]. For this purpose, a time-dependent MSP model with climatic covariates [28] as well as a spatial-temporal linear model [26,29] can be useful. Addressing these challenges may be the first step in delving further into this research area. Moreover, increased collaboration between climate scientists and statisticians is required.

**Supplementary Materials:** The following supporting information can be downloaded at: https://www.mdpi.com/article/10.3390/atmos13040589/s1.

**Author Contributions:** Conceptualization, P.B. and P.S.; methodology, P.B., J.-S.P. and P.S.; software, P.S. and T.P.; validation, P.G., J.-S.P. and P.B.; formal analysis, J.-S.P. and P.B.; investigation, P.B.; data curation, P.S. and T.P.; writing—original draft preparation, P.S. and T.P.; writing—review and editing, P.B., P.S. and T.P.; supervision, T.P.; project administration, P.S., J.-S.P. and P.B.; funding acquisition, P.B. All authors have read and agreed to the published version of the manuscript.

**Funding:** This research was financially supported by Mahasarakham University. Park and Thanawan's work was supported by the framework of the international cooperation program managed by the National Research Foundation of Korea (FY2021K2A9A1A01102193). Guayjarernpanishk's work was supported by Khon Kaen University.

**Institutional Review Board Statement:** Not applicable.

**Informed Consent Statement:** Not applicable.

**Data Availability Statement:** Not applicable.

**Acknowledgments:** Observation data in Thailand provided by the Thai Meteorological Department (TMD) at https://www.tmd.go.th. (accessed on 8 August 2021).

**Conflicts of Interest:** The authors declare no conflict of interest.

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
