# Peer review of "Spatial Modeling of Extreme Temperature in Northeast Thailand"

_atmosphere, doi:10.3390/atmos13040589_

Round 1

Reviewer 1 Report

In the manuscript, the authors attempted to investigate the extreme distribution and return level of daily maximum temperature in northeastern Thailand using both local GEV and spatial GEV methods. The scientific content may be of interest, but the language needs improvement. I recommend that the authors send the manuscript to a language editing service.

Author Response

Dear Reviewer,

Thank you to your comment.  We try our best to revise this manuscript. After we edited our manuscript we sent it to native speaker for grammatical validation which is expected to take about a week.

Best regards,

Thanawan Prahadchai

Reviewer 2 Report

The manuscript entitled ‘Spatial Modelling of Extreme Temperature in Northeast Thailand’ concerns quite an important methodological topic of spatial air temperature distribution on the example of NE Thailand. It is typical case study concerning meteorological and geostatistical aspects. My general opinion is positive however, there are some points which should be clarified and added. There are as follows:

In the article, I highly appreciate the method itself and it is the strength of the article. Perhaps it is worth applying this method to other areas, although it requires additional analyzes. Therefore, in the article it is worth indicating how much the proposed method is universal. Simply, to what other regions it can be applied, e.g. in other zones and types of climate. This also applies especially to those areas where the complexity of the terrain is significantly greater.

Only the main geographic variables (latitude, longitude, altitude) were analyzed in the study. Other variables were not taken into account, e.g. type of terrain, slope or land use, type of vegetation, etc. These parameters could be also quite important for the issue. Please explain and comment.Please provide more information concerning original data quality and homogeneity. It is crucial for the final results.

Line 106. Not quite clear: “annual daily maximum” please explain what values have been used.

Fig. 1. Quite interesting form of the figure but not quite readable. Please improve it slightly and put the name of the OX axis.

Fig. 3. It should be explained in the title that numbers denote station ID.

Some conclusions sound quite clearly while many scientists could be sceptic taking into account details as well as different meteorological fields. Please write some comments about the method when someone can apply it for other meteorological parameters like e.g. precipitation or wind speed. Please provide strengths and weaknesses of the method (advantages and disadvantages).

Author Response

Dear, Reviewer

"Please see that attachment"

Best regards,

Thanawan Prahadchai

Reviewer 3 Report

Dear Author,

Please consider the attached file for my comments.

All the best.

Author Response

(The authors gave the same response as above.)

Reviewer 4 Report

Comments are attached.

Author Response

(The authors gave the same response as above.)

Round 2

Reviewer 1 Report

Specific comments and edits:

1 Lines 1-3: “The goal of this work is to examine the annual maximum temperature in the northeast of Thailand by using data from 25 stations and employing spatial extreme modeling which is based on max-stable process (MSP) with schlatter’s method and predict the future temperature.”

=> “The goal of this work is to examine and predict the annual maximum temperature in the northeast of Thailand by using data from 25 stations and employing spatial extreme modeling which is based on max-stable process (MSP) with schlatter’s method”

2 Lines 5-7: “The study found that 15 stations are Weibull distribution and 10 stations

are Gumbel distribution, the significant Kormogolov-Smirnov (KS) test performs for all station that local GEV model is suitable.”

=> “The study found that 15 stations are Weibull distribution and 10 stations

are Gumbel distribution, with significant Kormogolov-Smirnov (KS) test performed for all station that local GEV model is suitable”

3 Line 11: “affect in” => “affect”

Author Response

Dear Reviewer,

Thank you for comment. We update new abstracts and edit them according to your recommendations.

Please check the latest modified version of our manuscript.

Best regards,

Thanawan Prahadchai

Reviewer 3 Report

Dear Authors,

Thank you for your responses to my comments. However, I wish you had not submit it before the English correction. As far as I checked your responses, I saw that you have not replied to some of my comments. Please check the comments below:

  • Comment 6 in previous round is not replied and is not done.
  • Comment 21 in previous round should be added to the text. If a reviewer commented on an issue, please do not explain him your response, instead he/she asks you to give it in the text.
  • Comment 22 in previous round is not done.

Next round, please submit the language edited revised version. If you need time, just let the journal office know.

All the best.

Author Response

Dear Reviewer,

Best regards,

Thanawan Prahadchai

Reviewer 4 Report

Dear Authors, thanks for the extra effort. I would like to recommend the publication of a new version of the manuscript.

Author Response

(The authors gave the same response as above.)
